# The Impact of Sport Education on Chinese Physical Education Majors' Volleyball Competence and Knowledge

Ping Li [1,†], Wei Wang [1,†], Hairui Liu [2,3], Chunhe Zhang [4,*] and Peter A. Hastie [3,*]

1. College of Physical Education, Hubei Normal University, Huangshi 435002, China; pedagogy2021@foxmail.com (P.L.); wangweipe2021@foxmail.com (W.W.)
2. Department of Kinesiology & Applied Health, The University of Winnipeg, Winnipeg, MB R3B 2E9, Canada; ha.liu@uwinnipeg.ca
3. School of Kinesiology, Auburn University School of Kinesiology, Auburn, AL 36849, USA
4. College of Physical Education, Hubei University, Wuhan 430061, China
* Correspondence: chunhez0615@foxmail.com (C.Z.); hastipe@auburn.edu (P.A.H.); Tel.: +1-334-844-1469 (P.A.H.)
† These authors contribute equally to this paper.

**Abstract:** The Sport Education curriculum model, while well studied in primary and secondary school settings, has been much less evaluated in university physical education. In this study, 110 Chinese university students were randomly assigned to participate in 6 classes taught using either Sport Education or a more traditional teacher-directed style. Data were collected on the students' skill execution, game performance, and knowledge. Over the course of a 16-week term, all participants showed significant improvements. However, the Sport Education students' gain scores were significantly higher after controlling for pretest scores. It is suggested that the features of Sport Education that have been shown to motivate students in previous studies (persisting teams, developmentally appropriate competition, and taking roles other than player) serve to stimulate students toward achieving the multiple goals of Chinese university physical education.

**Keywords:** sport education; volleyball; physical education; experimental design; China

## 1. Introduction

The Sport Education model was introduced to the physical education community in 1994 with the release of Daryl Siedentop's text *Sport Education: Quality PE Through Positive Sport Experiences.* The stimulus for the entire concept of Sport Education was Siedentop's belief that the way the sporting experience was presented to students in school physical education did not contain the elements he considered garnered students' engagement and enthusiasm in sport settings outside of school. In essence, he believed that sport in physical education had become "decontextualized" [1], offering an inauthentic experience that left students feeling bored and unchallenged.

In creating Sport Education, Siedentop focused on two key elements of the ways sport is organized in youth, school, and club sport: (i) the idea of the persisting team and (ii) meaningful, consequential competitions. These contests add meaning to the experience and subsequently are treated seriously and with high levels of affect [2]. As Siedentop et al. [3] commented, "the idea of the persisting team is one of the most fundamental and non-negotiable aspects of Sport Education. Whereas in most physical education settings, teams are formed only for the duration of a game, in Sport Education students not only play together but also practice skills, develop tactics, and complete administrative tasks as a team." Researchers have also observed that team affiliation is one of the most attractive features of the model [4–6].

Sport Education sees competition as fundamental to the sport experience. However, in contrast to the zero-sum gain idea held by many, competition within the model is defined as

the pursuit of excellence. The contest becomes with oneself to better a previous standard or performance. For this to be authentic, however, testing oneself against another (individual or team) is seen as essential. Sport Education therefore encourages competition, not against a particular adversary, but as a process of discovery [3].

In the educative wing of Sport Education, Siedentop followed two strategies. First, he took the good and attractive parts of "out of school sport" and included them in physical education. Next, he modified those features that had the potential to marginalize and alienate students by introducing the idea of developmentally appropriate competition through the use of small-sized teams, modified game forms, graded competition, as well as the requirement that all players participate all the time.

In an effort to expand beyond the role of "player", Siedentop introduced the idea that students would not only participate in games and contests but would also take up roles and responsibilities within their team, either related to internal team tasks or as officials. His final goal was that all of these experiences would be bound together by the notion of festivity, where there was an inbuilt structure, a celebratory nature of participation, etc.

Since the first release of the 1994 book, there have been three subsequent editions [3,7,8] that have used the evidence from practitioners and researchers to present evidence for the efficacy of the model, to consolidate the essential premises and features of the model that lead to successful implementation, and to expand the original model in numerous ways and possibilities. The growing research base and updated model have made Sport Education the most studied and published curriculum model in physical education. Major reviews have been written on the demographics of adoption, the responses of teachers and students, and the ability of the model to achieve its goals. Early reviews were of descriptive studies that then progressed to more quasi-experimental designs and, as of this writing, full experiments, extensive case studies, and some longitudinal work. The paper by Hastie and Wallhead [9] provides a concise history of this research development.

Researchers investigating Sport Education originally focused on its ability to achieve the five "big aims" of PE, namely skill development, knowledge and understanding, fitness, social development, and values/attitudes [10]. The most recent work has focused on Sport Education's ability to develop young people in schools as competent, literate, and enthusiastic sports players. Running parallel to these research efforts, projects have aimed at understanding how Sport Education can best be presented to teachers (both pre-service and in-service) to train them to implement the model. Best practices have been identified, and the dilemmas that teachers face have also been described [11–13]. One area in which research on Sport Education is still limited is higher education. Initial descriptive studies showed that students had perceptions of greater group cohesion and team affiliation [14,15] and that the model was effective in developing their skills and knowledge [16,17]. More recently, the research has shifted to the potential of Sport Education to promote physical activity outcomes [18,19]. At this point, however, the only identifiable report within the realm of Sport Education's foundational objectives (competence, literacy, and enthusiasm) is the study of Pritchard et al. [20], which showed significant improvements in badminton game performance and knowledge. It should be noted that this study did not involve a control group and was limited to one class.

In comparison to the studies in Sport Education mentioned above, the current study reaches "state of the art" status in two ways. First, from an "ideas" perspective, it expands the study of a physical education curriculum within Chinese universities in line with new government directives. Second, from a "methodological" perspective, it is the first study of Sport Education in any setting that randomly assigned participants to the experimental groups.

The purpose of this study was to use a pre–post experimental design to examine the impact of Sport Education on Chinese physical education majors' volleyball competence and knowledge. With the advent of Chinese curriculum reform to promote more active engagement and the development and cultivation of students' individual abilities, several papers in Chinese journals have also espoused the possibilities of a good fit between the

goals of the Sport Education pedagogical model and the new guidelines [21,22]. The argument has been that Sport Education involves an authentic sport experience and focuses on independent and cooperative student learning, whereas traditional physical education is more whole group health-focused with significantly less opportunity for social engagement [22]. Consequently, studies of Sport Education in China have reported successful implementation in both Chinese and English journals. Outcomes have included increases in students' intrinsic motivation and task orientation [23], enhanced student enthusiasm and initiative, and improved interest in learning and participation [24] as well as students' perceptions of increases in motor skills [25,26]. For this study, it is hypothesized that students in volleyball classes conducted using Sport Education will have significantly greater gains in their skill levels, game performance, and volleyball knowledge than their classmates who participated in classes conducted in a more teacher-directed, skill-focused form of instruction.

## 2. Materials and Methods

### 2.1. Participants

One hundred ten university physical education majors (80 males: mean age $19.42 \pm 0.73$) from 6 classes of a state university in central China were the participants in this study (see Table 1). Students were registered for the volleyball class as a whole (rather than a specific class), allowing for participants to be randomly assigned to two groups (Traditional Instruction or Sport Education). The Traditional Instruction group consisted of 3 classes of 55 students (39 males), while a further 3 classes of 55 students participated in the Sport Education group (41 males).

**Table 1.** Participant demographics.

| Class | Male (*n*) | Female (*n*) | M $\pm$ sd of Age |
|---|---|---|---|
| Sport Education 1 | 21 | 0 | $19.50 \pm 0.58$ |
| Sport Education 2 | 20 | 0 | $19.33 \pm 0.58$ |
| Sport Education 3 | 0 | 14 | $19.00 \pm 0.39$ |
| Traditional Instruction 1 | 19 | 0 | $19.42 \pm 0.84$ |
| Traditional Instruction 2 | 20 | 0 | $19.70 \pm 0.86$ |
| Traditional Instruction 3 | 0 | 16 | $19.31 \pm 0.70$ |

One male faculty member taught all classes. He had taught volleyball for more than 20 years to both physical education majors and students from the general population. The instructor also had experience teaching a number of volleyball classes incorporating Sport Education, having conducted seasons prior to this study.

### 2.2. Class Designs

The students participated in 2, 90 min lessons per week over 16 weeks. Common content across the Traditional Instruction and Sport Education conditions included (a) skill performance of serve, forearm pass, set and spike; (b) instruction in the basic tactics to play successfully in 5 versus 5 games; and (c) participation in games. Table 2 provides an outline of the progressions of technical skills covered for both classes.

For the Traditional Instruction group, the focus was wholly on learning the fundamental technical skills and tactics and then applying them in non-modified games. Lessons consisted of 10 min of physical training and warmup, followed by 60 min of teacher demonstration, individual skill and partner practice, and concluded with 15–20 min of game play.

For the Sport Education group, the season consisted of three phases across the 16-week semester. The first phase (5 weeks) focused on teaching key skills, introducing basic tactical strategies, and learning the rules and regulations of volleyball. After the first 5 weeks, the students in each class were divided into 4–5 equally skilled teams. Each team was responsible for choosing its name, color, and mascot before progressing to team practice. Each team member took one role—either captain, coach, strength and conditioning coach,

equipment manager, or team statistician—in addition to the officiating roles of referee, scorekeeper, or statistician that are common in Sport Education. This second phase (5 weeks) involved team practice as well as informal (3 vs. 3) games between teams. These games were organized so that each team had the opportunity to become familiar with the organization of the competition and to practice officiating responsibilities such as referee or scorekeeper. The third phase (6 weeks) saw teams engaged in 5 vs. 5 matches using a 21-point scoring system and best two out of three sets. A final championship game was held during week 16 that was followed by a ceremony to end the whole season.

**Table 2.** Progressions of technical skills covered for both classes.

| Phase/Week | | Sport Education | Traditional Instruction |
|---|---|---|---|
| Skill Instruction | 1 | Positions, self-passing, two hand passing, underhand serve | Positions, self-passing, shuttle run |
| | 2 | Steps, two hand passing, spike, review passing and underhand serve | Steps, two hand toss passing |
| | 3 | Spike, stand toss serve, paired passing | Review steps, self-passing, toss passing |
| | 4 | Block, over net passing, setting | Over net passing, underhand serve |
| | 5 | Offense, defense, receiving | Review underhand serve, two hand toss passing, strength and conditioning practice |
| First Volleyball Season 3 vs. 3 | 6 | Team practice and 3 vs. 3 demonstration | Spike, stand toss serve, paired passing |
| | 7 | Team practice and preseason (3 vs. 3) | Review spike, standing toss serve, paired passing |
| | 8 | Team practice and preseason (3 vs. 3) | Setting, review spike and toss serve |
| | 9 | 3 vs. 3 game season | Block, review serve and receiving |
| | 10 | 3 vs. 3 game season | Review serve and receiving. Offense and defense positions |
| Second Volleyball Season 5 vs. 5 | 11 | Team practice and 5 vs. 5 demonstration | Review offense and defense positions |
| | 12 | Team practice and preseason (5 vs. 5) | Rules and regulations. Volleyball tactics |
| | 13 | Team practice and preseason (5 vs. 5) | Games |
| | 14 | 5 vs. 5 game season | Games |
| | 15 | 5 vs. 5 game season | Games |
| Final Week | 16 | Final Championship and Ceremony | Skill Tests |

### 2.3. Fidelity of Condition

The instruction checklist developed by Pritchard et al. [27] was used to examine the characteristics of each condition. Members of the author team not involved in the actual teaching of the classes checked both lesson plans and videotapes of lessons showing both models. The observers coded items of 4 randomly selected lessons from each of the six classes and reached a 100% agreement regarding the instructional approach used in each.

### 2.4. Measures

The following data were collected for this study: (a) students' performance on volleyball-specific skills (forearm pass and overhead pass); (b) students' volleyball game performance; and (c) students' volleyball content knowledge. All assessments were administered in the week immediately prior to the commencement of the course and in the week following the game season (before final examinations).

### 2.4.1. Game-like Skills

Two functional skill tests (forearm passing and overhead setting) developed by Byra [28] were used to assess the students' ability to execute the critical volleyball skills of the forearm pass and of setting. Table 3 provides a description of all the tests.

**Table 3.** Description of game-like skill tests.

| Test | Task | Scoring | Points |
|------|------|---------|--------|
| Game-like setting | Thrower tosses an underhand pass to a setter who attempts to pass it through a hoop at net height on the sideline. | 10 trials<br>2 = through the hoop<br>1 = touches the hoop<br>0 = misses the hoop | 20 |
| Game-like passing | Thrower tosses an underhand pass to a receiver who attempts to make a forearm pass to a catcher standing in a 2 m² target area at the net. | 10 trials (5 on two positions on the court)<br>2 = catcher catches the ball in the square and above the head<br>0 = below the head or outside the area | 20 |

### 2.4.2. Game Performance

The systematic observation system developed by Mesquita et al. [29] was used to assess students' game performance, with all students participating in two 21-point matches, one pre-intervention, the other post-intervention. All matches were captured using a GoPro HERO7 camera placed in the middle of the court and above the net at a height of approximately 4.5 m to capture the whole volleyball court. Each in-game action made by players was judged as either "present" or "absent" on four variables: adjustment, decision making, skill technique, and skill outcome.

The second and fourth authors coded all game performance data, and only data for which the percentage of agreement between these two coders exceeded 80% were included in the analysis. A two-step process was used to measure the reliability of the coders. First, four games were randomly selected from the pre and posttest, which included samples of eight volleyball games and 32 participants. These games were used as training films, where the observers discussed and agreed on their understanding of the four variables. Second, the observers practiced collaborative and independent coding to a point where they reached a greater than 80% agreement across all eight games. For this study, the final agreements reached 96% for adjustment, 92% for decision making, 95% for skill technique, and 98% for outcome.

### 2.4.3. Volleyball Knowledge

One week before and after the study, each participant completed a validated 40-item volleyball content knowledge test that included knowledge of rules and etiquette as well as knowledge of techniques and tactics [30].

### 2.5. Data Analysis

Scatter plots among the dependent variables were plotted to identify any potential outliers visually. Normality and homogeneous variances were also checked before further analysis. Paired-sample t-tests were conducted to measure the gain scores for the four dependent measures across time for the two groups. Effect size was measured used Cohen's d.

Four separate one-way analyses of covariance (ANCOVA) were conducted to determine any statistical differences between the effects of Sport Education and Traditional Instruction on post-intervention scores for forearm pass, overhead pass, game performance,

and knowledge, after controlling for pre-intervention scores. Statistical analyses were performed with the SPSS 25 (SPSS Inc./IBM, Armonk, NY, USA). Effect sizes of <0.2, 0.2–0.5, 0.5–0.8, and >0.8 represented extremely low, low, medium, and high effect sizes, respectively [31].

### 3. Results

Visual inspection of the scatter plots did not find any outliers in the data set, and the homogeneous variances test of all pretests and posttests showed no violations of assumptions. All data met normality and normal distribution assumptions without statistical violations.

The results of the students' performance on the four dependent measures are presented in Table 4. With the exception of passing in the Traditional Instruction group, students in both units demonstrated statistically significant improvements from pretest to posttest for all volleyball activities. Analysis of the effect sizes, however, provided evidence to suggest Sport Education is a more effective tool for enhancing students' technical performance within these events. All effect sizes for Sport Education classes were classified as "high", while those for Traditional Instruction (with the except of Game performance) were either medium or low.

**Table 4.** Gain scores for all measured variables by group.

| Test | Group | Pre-Test M (SD) | Post-Test M (SD) | t | p | d |
|------|-------|-----------------|------------------|---|---|---|
| Passing | Sport Education | 3.89 (3.54) | 8.15 (4.03) | 6.62 | <0.001 | 1.12 |
| | Traditional Instruction | 4.04 (3.34) | 4.93 (3.33) | 1.65 | 0.105 | 0.27 |
| Setting | Sport Education | 2.73 (3.01) | 6.11 (3.44) | 7.73 | <0.001 | 1.04 |
| | Traditional Instruction | 2.45 (2.24) | 4.22 (2.83) | 4.36 | <0.001 | 0.69 |
| Game performance | Sport Education | 0.36 (0.11) | 0.54 (0.11) | 10.02 | <0.001 | 1.72 |
| | Traditional Instruction | 0.36 (0.10) | 0.44 (0.12) | 3.98 | <0.001 | 0.66 |
| Content knowledge | Sport Education | 21.76 (4.06) | 31.36 (4.97) | 7.82 | <0.001 | 2.12 |
| | Traditional Instruction | 19.04 (5.39) | 26.27 (4.18) | 5.44 | <.0001 | 1.49 |

### 3.1. Passing Skills

Figure 1 shows the mean difference scores obtained in the two groups for the forearm pass. After controlling for pre-intervention scores (3.96), a statistically significant difference in posttest scores between the instructional groups was shown, F (1, 107) = 19.84, $p < 0.001$, partial $\eta^2 = 0.156$.

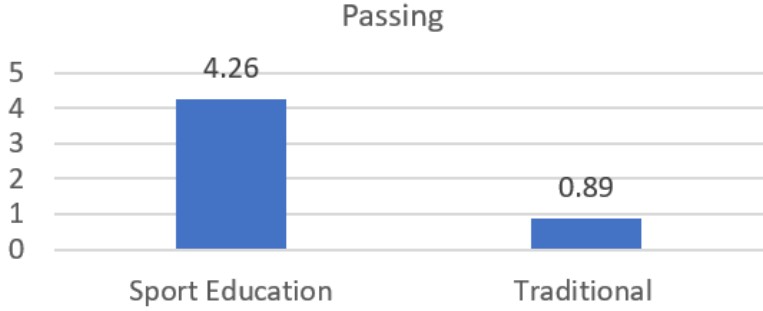

**Figure 1.** Mean difference scores for forearm passing.

### 3.2. Setting

Figure 2 shows the mean difference scores obtained in the two groups for overhead setting. After controlling for pre-intervention scores (2.59), a statistically significant difference in posttest scores between the instructional groups was shown, F (1, 107) = 19.84, $p < 0.001$, partial $\eta^2 = 0.156$.

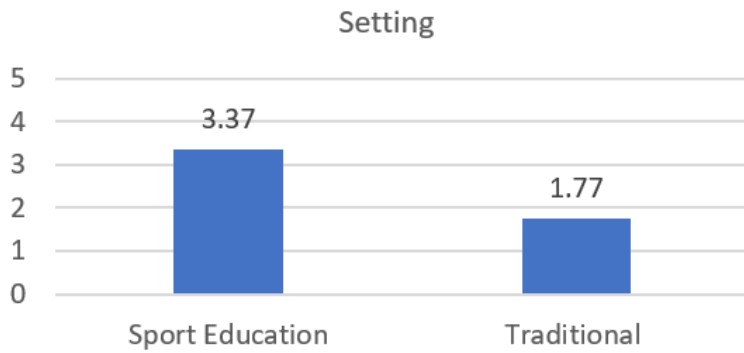

**Figure 2.** Mean difference scores for setting.

*3.3. Game Performance*

Figure 3 shows the mean difference scores obtained in the two groups for game performance. After controlling for pre-intervention game performance scores (0.36), a statistically significant difference in posttest scores between the instructional groups was shown, F (1, 107) = 19.84, $p < 0.001$, partial $\eta^2 = 0.156$.

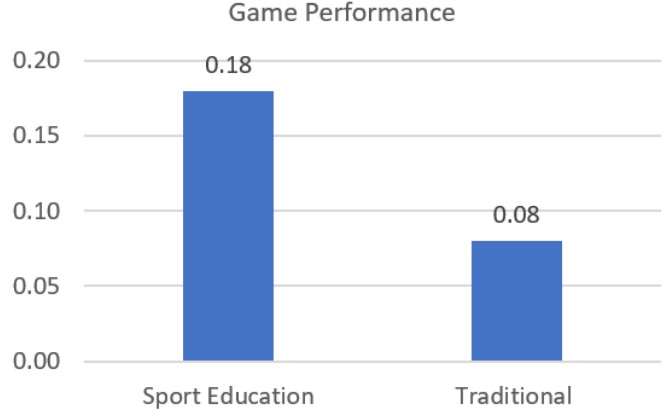

**Figure 3.** Mean difference scores for game performance.

*3.4. Content Knowledge*

Figure 4 shows the mean difference scores obtained in the two groups for CK. After controlling for pre-intervention CK scores (20.40), a statistically significant difference in posttest scores between the instructional groups was shown, F (1, 109) = 30.66, $p < 0.001$, partial $\eta^2 = 0.223$.

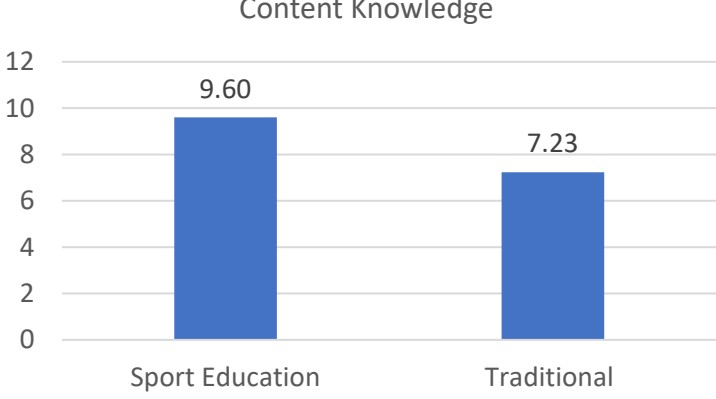

**Figure 4.** Mean difference scores for content knowledge.

## 4. Discussion

This study is the first to use an experimental design in research on Sport Education in which participants were randomly distributed into groups. Its goal was to examine the impact of the model on university students' attainment of volleyball competence and content knowledge. As an executive summary, students in all classes achieved statistically significant gain scores on all measures (with the exception of passing for the Traditional Instruction group), but students in the Sport Education group showed significantly greater improvement across the course.

### 4.1. Game Performance

Early studies of Sport Education suggested that the reason for the success of the model in achieving gains in students' skill and game performance was that they engaged in the content over a longer period of time than is typical within physical education programs [32–34]. However, a limitation of these studies was that they failed to include a control group. When this element was introduced into research on the model, it was found that students in both groups made significant gains [35–37]. Nonetheless, in all cases, the Sport Education students significantly outperformed their classmates who participated in a more traditional, teacher-directed instructional format. This study is consistent with those findings.

The underlying message here seems to be that, provided there is sufficient time for practice, all students can make gains in their game performance (irrespective of instructional format), a notion presented particularly early by Rink et al. [38]. More specifically, Miller [39] has concluded that an intervention volume greater than eight hours, or ten sessions, appears to be a common cut point for significant, positive changes in performance. That threshold was certainly achieved in this study.

Given these findings, we need to look for other factors that explain the differentiation between the Sport Education classes and those following Traditional Instruction. The three founding principles of Sport Education were the use of developmentally appropriate competition through the use of modified games, the taking of roles other than player, and the requirement that all players be involved all the time [1].

In this particular study, while the game of volleyball was not changed, the Sport Education students played in a 4 vs. 4 format versus the full-sided 6 vs. 6 format in the Traditional Instruction group. The argument for reducing the number of players is that it enhances not only the playing volume but also specific technical actions of the game [40,41]. Furthermore, it has been suggested that reducing the number of players creates a positive environment that fosters decision making, and therefore, more passes and skilled actions are performed correctly [42,43]. While the exact number of contacts with the ball were not measured in this study, it would be reasonable to expect that this would be the case.

Another popular postulate for the greater gains made by students during Sport Education is that competition is more authentic. In particular, games count towards determining the overall season champion, which leads to Sport Education being seen as more serious than regular physical education and students believing that the formal nature of the competition motivates them to train better. Indeed, this aspect has been seen in earlier studies of Sport Education in Chinese university physical education classes [25]. Team practices have been seen as more serious, and students report being more likely to pay attention to feedback given by the teacher and their teammates than might be the case during a more traditional teaching format.

### 4.2. Skill Development

As with game performance, the execution of volleyball skills showed improvements for both cohorts but with a significantly greater gain scores for the Sport Education students. In previous volleyball studies, this has been less that case [26]. We suggest that perhaps this has been due to the way in which skill has been measured. In this study, the skill tests were game-like tasks, in that they mirrored specific actions required during games. Although

this study did not measure issues of group cohesion, there is evidence from previous work with university students that the affiliation and persisting team membership inherent in Sport Education is a strong driver of the seriousness toward the contexts [14,16].

In Chinese physical education settings, we have found that instructors have reported higher levels of engagement by students in Sport Education contexts [26]. While recognizing that engagement is a multidimensional construct containing behavioral, cognitive and emotion, and affective elements [44], we would suggest that even if behavioral engagement levels were similar across groups, the cognitive and emotional aspects would be expected to be higher in Sport Education situations. These are elements relating to self-regulation and investment (cognitive) and positive attitude and interest (emotional) aspects.

*4.3. Knowledge*

In terms of knowledge, a number of studies have suggested that student participation in officiating roles in Sport Education has an impact on understanding the declarative and procedure knowledge of the sports they play [20,33,35,36]. In this study, it was determined that, given sufficient time, students would make improvements. However, the difference between Sport Education and Traditional Instruction is that students have to put this knowledge to use during their officiating roles (rather than as a recall exercise at the end of the course). It has been demonstrated that university students taking these roles show particularly high levels of active engagement [45]. By "active", it is meant that the student keeps up with the ball, follows play, consistently enforces the rules, and uses the whistle definitively.

In addition to playing games, however, students during Sport Education are also active observers in their roles as officials. It may be that by not only playing games, but also watching other students in a focused way, helps promote tactical awareness and reinforces content knowledge [33,35]. That is, as an official, students may be thinking along the lines of "if I were on the court, where would I have hit the ball", or perhaps in some cases, "what decision would I have made"?

**5. Conclusions**

This is the first study involving the Sport Education model to utilize a true randomized control design. Across all measures, the students in Sport Education classes outperformed their classmates who followed the Traditional Instructional format of direct teacher instruction and a focus on sport techniques. The explanation for these findings mirrors those from previous research in university and school settings. That is, the features of Sport Education that have been shown to motivate students in previous studies (persisting teams, developmentally appropriate competition, and taking roles other than player) serve to prompt students toward achieving the multiple goals of Chinese university physical education. These results are practically significant given the new focus of Chinese physical education on developing students' individual abilities and knowledge [21,22].

**Author Contributions:** P.A.H. conceived the study, participated in its design, performed inferential analysis and wrote the final manuscript. P.L. and W.W. conducted all the instruction and supervised the collection of all data. H.L. acted as project manager and was responsible for all coder training and reliability calculations. C.Z. video recorded all instructional sessions and game performance data collection and led the coding of game performance. All authors have read and agreed to the published version of the manuscript.

**Funding:** This research received funding from Hubei Province Higher Education Provincial Teaching Research Project: A practical study on innovation of college physical education integration—a systematic science theory perspective, No. 2020612.

**Institutional Review Board Statement:** The study was conducted according to the guidelines of the Declaration of Helsinki and approved by the Institutional Review Board of Hubei Normal University (protocol code 2020069).

**Informed Consent Statement:** Informed consent was obtained from all subjects involved in the study.

**Data Availability Statement:** The data presented in this study are available on request from the corresponding author. The data are not publicly available due to privacy restrictions.

**Conflicts of Interest:** The authors declare no conflict of interest.

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
