# Peer review of "The Impact of Sport Education on Chinese Physical Education Majors’ Volleyball Competence and Knowledge"

_sustainability, doi:10.3390/su14031187_

Round 1

Reviewer 1 Report

This study hypothesized that volleyball classes conducted using Sport Education will improve skill levels, game performance, and volleyball knowledge than their classmates who participated in classes conducted in a more teacher-directed. Thus, the authors conducted a experiment that test the difference between these pedagogical approaches. In this sense, I have some general and specific concerns.

General

The authors write a gap and point out (what I understand) the "state of the art" on research involving sports education. However, only “still limited is higher education” does not justify the article. Additionally, the summary results of anterior papers were presented. In this sense, what is the difference between these and the present paper, and what does your article add to Sport Education theory?

A central concern in this paper it’s about the method of randomly. How were participants randomized? Are they blind? The instructor was not blind, given that he taught all classes. This is a serious bias risk. How do the authors deal to minimize this? 

Specific comments

Abstract

Pg 1; L-16: a comma (“,”) after  game performance

Pg 1; L-17-18: I missed specific results (mean, sd, effect size, p, etc)

Introduction

Pg 2; L: 35-36: Please, provide a peer review reference to this affirmation.

  1. Materials and Methods

Pg 3; L-113: please, provide the mean age of females. What do you mean  with intact class?

Pg 3; L-124: Please, present the schedule of 24 sessions over 16 weeks Ex: week one. Two sessions week two one session...

Data Analysis

Pg 3; L-197: Please, present the cut off effect size.

Results

               Please, clarify the same pre-intervention scores (0.36) to three different results.

               Why the results of baseline are the same to Traditional and Sport Education in figures?

Author Response

This study hypothesized that volleyball classes conducted using Sport Education will improve skill levels, game performance, and volleyball knowledge than their classmates who participated in classes conducted in a more teacher-directed. Thus, the authors conducted a experiment that test the difference between these pedagogical approaches. In this sense, I have some general and specific concerns

...........

General

The authors write a gap and point out (what I understand) the "state of the art" on research involving sports education. However, only “still limited is higher education” does not justify the article. Additionally, the summary results of anterior papers were presented. In this sense, what is the difference between these and the present paper, and what does your article add to Sport Education theory?

** We have added the following:

With reference to the studies in Sport Education mentioned above, the current study reaches “state of the art” status in two ways. First, from an “ideas” perspective, it expands the study of physical education curriculum within Chinese universities in line with new governmental directives. Second, from a “methodological” perspective, it is the first study of Sport Education in any setting that randomly assigned participants to the experimental conditions.

A central concern in this paper it’s about the method of randomly. How were participants randomized? Are they blind? The instructor was not blind, given that he taught all classes. This is a serious bias risk. How do the authors deal to minimize this? 

** The participants were randomly designed by sport faculty at the university who were not part of the study and were blind to its objectives.

While the instructor was not blinded to the study, the advantage of having a single instructor is that any teacher affects are eliminated. 

.............

Specific comments

Abstract

Pg 1; L-16: a comma (“,”) after  game performance

** added

Pg 1; L-17-18: I missed specific results (mean, sd, effect size, p, etc)

** All these data are available in Table 4 and the notes in the results. To place them in the abstract would exceed the word limit, as there are a large number of dependent measures.

Introduction

Pg 2; L: 35-36: Please, provide a peer review reference to this affirmation.

** added

  1. Materials and Methods

Pg 3; L-113: please, provide the mean age of females. What do you mean  with intact class?

** These are included in the addition of Table 1

** intact has been deleted.

Pg 3; L-124: Please, present the schedule of 24 sessions over 16 weeks Ex: week one. Two sessions week two one session...

** should be 2 sessions, not 24. sorry

Data Analysis

Pg 3; L-197: Please, present the cut off effect size.

** These are included as a note in Table 4

Results

Please, clarify the same pre-intervention scores (0.36) to three different results.

** The game performance scores of both groups were the same at pre-test

Why the results of baseline are the same to Traditional and Sport Education in figures?

** Originally, we plotted the adjusted pre-test scores to show the strength of the covariance results. However, in keeping with another reviewers recommendation of showing the gain scores, these figures have been changed.

Reviewer 2 Report

This paper deals with the  impact of sport education on Chinese physical education majors’ volleyball competence and knowledge

Main Concerns:

  • a state of the art section is missing, thus, it is impossible to assess the novelty of the proposed paper;

ABSTRACT

  • The abstract provides a brief description of the context, the proposed research, and the obtained results.

The introduction provides a very brief overview of the research domain.

Main Concerns:

  • the authors should provide more details to highlight the relevance of the proposed paper and to describe in a comprehensive way the problem they try to address;
  • an analysis of the state of the art is missing.

The authors should add the section Results, which is usually adopted for the findings section of a paper.

The paper should be divided into the following sections: 1. Introduction, 2. Materials and Methods, 3. Results, 4. Discussion, 5. Conclusions

The paper needs to be better reorganized also in subsections, with better and easier differentiation.

Authors should take into account more previous works (e.g. theoretical, conceptual, and empirical reviews) published in the literature. Authors should discuss the results and how they can be interpreted from the perspective of previously published studies. I suggest adding a reference: Fedushko S., Davidekova M. Analytical service for processing behavioral, psychological, and communicative features in online communication.  Procedia Computer Science. Volume 160, 2019, Pages 509-514. https://doi.org/10.1016/j.procs.2019.11.056

I suggest adding a concluding paragraph with that, how these main findings of the paper address the challenge of sustainability.

Thank you for a good job.

Author Response

This paper deals with the impact of sport education on Chinese physical education majors’ volleyball competence and knowledge

Main Concerns:

  • A state of the art section is missing, thus, it is impossible to assess the novelty of the proposed paper;

ABSTRACT

  • The abstract provides a brief description of the context, the proposed research, and the obtained results.

** We have tried to provide all the key information here for readers.

  • The introduction provides a very brief overview of the research domain.

Main Concerns:

  • The authors should provide more details to highlight the relevance of the proposed paper and to describe in a comprehensive way the problem they try to address
  • An analysis of the state of the art is missing.
  • The authors should add the section Results, which is usually adopted for the findings section of a paper.

** The results section appears on page 6

  • The paper should be divided into the following sections: 1. Introduction, 2. Materials and Methods, 3. Results, 4. Discussion, 5. Conclusions

** These are indeed the sections included in the first submission,

The paper needs to be better reorganized also in subsections, with better and easier differentiation.

** The protocols for the journal were followed. All sections and subsections are numbered and labeled.

  • Authors should take into account more previous works (e.g. theoretical, conceptual, and empirical reviews) published in the literature. Authors should discuss the results and how they can be interpreted from the perspective of previously published studies. I suggest adding a reference: Fedushko S., Davidekova M. Analytical service for processing behavioral, psychological, and communicative features in online communication.  Procedia Computer Science. Volume 160, 2019, Pages 509-514. https://doi.org/10.1016/j.procs.2019.11.05

** We are not sure how this reference is applied to the study presented here.

  • I suggest adding a concluding paragraph with that, how these main findings of the paper address the challenge of sustainability.

** A section has been added.

  • Thank you for a good job.

** We appreciate he kind words and the helpful comments

Reviewer 3 Report

I find this topic interesting and important. Physical education at the University level is scarce and insufficient and this space should be better defined and advocated for, as it is an important pillar of sustainable health prevention. I found the paper well written and easy to read.

Methods

Explain in more details the process of participant recruitment.

Given that you state you checked the normality and homogeneity, you could provide the statement on whether the date were normally distributed.

Which statistical software did you use for the analyses? What significance level did you use?

Results

Figure 1 although provides visual representation in trends, does not provide information that does not exist already in previous text. It shows the mean values that are already presented in Table above. Also, it does not provide information non significance nor effect sizes so it could not be used as stand-alone figure. I suggest removing this Figure.

Figures 2, 3, 4 – check the previous comment. It would be nice to have visual representation of results but something that is not already in the table above in more detail. As it is now, figures do not provide anything new and could be only a distraction. For instance you could plot the mean difference (effect) obtained in traditional and sport education, and show the whether the effect of sport education was larger and to what degree compare to traditional approach.

Discussion

Add limitations section.

The first paragraph of conclusion is too general. Add another that encompass practical implications and what could be possible outcomes of these practical implications.

Author Response

I find this topic interesting and important. Physical education at the University level is scarce and insufficient and this space should be better defined and advocated for, as it is an important pillar of sustainable health prevention. I found the paper well written and easy to read.

** We appreciate the kind words and encouragement

Methods

Explain in more details the process of participant recruitment.

** There actually was no recruitment per se. The sample included all students who were registered for the courses that we taught.

Given that you state you checked the normality and homogeneity, you could provide the statement on whether the date were normally distributed.

** This section is added in the beginning of the results. It reads as follows.

Visual inspection of the scatter plots did not find any outliers in the data set, and the homogeneous variances test of all pretests and posttests showed no violations of assumptions. All data met normality and normal distribution assumptions without statistical violations.

Which statistical software did you use for the analyses? What significance level did you use?

** Have added: Statistical analyses were performed with the SPSS 25 (SPSS Inc./IBM, Armonk, NY, USA).

Results

Figure 1 although provides visual representation in trends, does not provide information that does not exist already in previous text. It shows the mean values that are already presented in Table above. Also, it does not provide information non significance nor effect sizes so it could not be used as stand-alone figure. I suggest removing this Figure.

Figures 2, 3, 4 – check the previous comment. It would be nice to have visual representation of results but something that is not already in the table above in more detail. As it is now, figures do not provide anything new and could be only a distraction. For instance you could plot the mean difference (effect) obtained in traditional and sport education, and show the whether the effect of sport education was larger and to what degree compare to traditional approach.

** All 4 figures have been replaced by those as your recommended… showing the mean gain scores

 Discussion

Add limitations section.

The first paragraph of conclusion is too general. Add another that encompass practical implications and what could be possible outcomes of these practical implications.

** a section has been added following this paragraph that reads: These results are practically significant given the new focus of Chinese physical education on developing students’ individual abilities and knowledge [20,21]

Reviewer 4 Report

First of all, I would like to congratulate the authors for their work on the impact of sports education on competition and knowledge of Chinese volleyball specialties in physical education.

The abstract is correct and provides sufficient information about the article.

The introduction has important theoretical references from the last five of the last five years, so it is up to date and well constructed.

In relation to the empirical framework, the sample is well defined, it is suggested to characterize the sample a little more if possible and to include a table to make the interpretation of the data more visual.

The conclusions are correct. Congratulations. Finally, this is a solid paper that adds to a field of sports science research.

Author Response

First of all, I would like to congratulate the authors for their work on the impact of sports education on competition and knowledge of Chinese volleyball specialties in physical education.

** Thank you. We have added a small piece in response to R1 that further highlights the key nature of the uniqueness of this paper.

The abstract is correct and provides sufficient information about the article.

** We have tried to present the best possible set of information in the space allowed.

The introduction has important theoretical references from the last five of the last five years, so it is up to date and well constructed.

** This study of multiple levels of knowledge and performance is still new, so we have relied on the most recent findings.

In relation to the empirical framework, the sample is well defined, it is suggested to characterize the sample a little more if possible and to include a table to make the interpretation of the data more visual.

** This has been added.

The conclusions are correct. Congratulations. Finally, this is a solid paper that adds to a field of sports science research.

** Thanks again.

Round 2

Reviewer 1 Report

The authors present a significant improvement in the manuscript.
However, serious concerns stayed.

per definition, this is a quasi-experimental or pre post experimental (see O'donoghue, P. (2009). Research methods for sports performance analysis. Routledge.)

Pg 1; L: 35-36: Please, provide a "peer review" reference, not the same book cited above. 

Pg 3; L 136: only two sessions? In 16 weeks? Please, clarify. It is intriguing that in two lessons there was technical and knowledge improvement. In which weeks were these lessons?

Table 1: Legend to SE, TE, Ave? Did you mean m +- sd of age?

Pg 7; L 222: this information here does not add anything and should be in the method.

Additionally, discuss in line with the results, the found effect sizes.

Pg 7; L: 233: Setting : After controlling for pre-intervention scores (0.36),

Pg 8; L: 240: Game performance: After controlling for pre-intervention game performance scores (0.36)

Were the same scores to setting and game performance?

Pg: 9; L: 255: what a "true" experimental design?

Pg 9; L: 272-279:  Sorry, I'm confused. Two lessons of 90 minutes it's far away from eight hours.

Pg 10; L: 321 A number of studies... The authors cities two manuscript (one published in 1990)

Author Response

Sustainability-1509191: Reviewer 1 Report, Round 2

  1. Per definition, this is a quasi-experimental or pre post experimental (see O'Donoghue, P. (2009). Research methods for sports performance analysis. Routledge.)

 ** we add this on L 99

  1. Pg 1; L: 35-36: Please, provide a "peer review"reference, not the same book cited above. 

** we added the following reference:

Sinelnikov OA., Hastie, PA. A motivational analysis of a season of sport education. Phys Educ Sport Pedagogy, 2010, 10(1), 55-69. https://doi.org/10.1080/17408980902729362.

  1. Pg 3; L 136: only two sessions? In 16 weeks? Please, clarify. It is intriguing that in two lessons there was technical and knowledge improvement. In which weeks were these lessons?

** we apologize for the confusion. There were 2 lessons each week over these 16 weeks. It is changed on L 133

  1. Table 1: Legend to SE, TE, Ave? Did you mean m +- sd of age?

** we added the full term in the first column, and changed the final column M ± sd of age. It is changed on L 131

  1. Pg 7; L 222: this information here does not add anything and should be in the method.

** This has been moved to L 208

  1. Additionally, discuss in line with the results, the found effect sizes.

 ** we placed the following sentence in the text.

All effect sizes for Sport Education classes were classified as “high”, while those for Traditional Instruction (with the except of Game performance) were either medium or low.” L 221

  1. Pg 7; L: 233: Setting : After controlling for pre-intervention scores (0.36),

** this was an error. It has been changed to the correct score of 2.59 L 237

  1. Pg 8; L: 240: Game performance: After controlling for pre-intervention game performance scores (0.36) Were the same scores to setting and game performance?

** this is the correct score (it was setting that was wrong)

  1. Pg: 9; L: 255: what a "true" experimental design?

** “true” has been deleted L 260

  1. Pg 9; L: 272-279:  Sorry, I'm confused. Two lessons of 90 minutes it's far away from eight hours.

** as noted on L 133 there were two lessons per week, so it far exceeded 8 hours.

  1. Pg 10; L: 321 A number of studies... The authors cities two manuscript (one published in 1990)

** the 1990 reference was incorrectly numbered. There are now 4 references to support this claim.   L 332